# Mechanical properties of rubble pile asteroids (Dimorphos, Itokawa, Ryugu, and Bennu) through surface boulder morphological analysis

Colas Q. Robin [1] ✉, Alexia Duchene [1], Naomi Murdoch [1], Jean-Baptiste Vincent [2], Alice Lucchetti [3], Maurizio Pajola [3], Carolyn M. Ernst [4], R. Terik Daly [4], Olivier S. Barnouin [4], Sabina D. Raducan [5], Patrick Michel [6,7], Masatochi Hirabayashi [8], Alexander Stott [1], Gabriela Cuervo[1], Erica R. Jawin[9], Josep M. Trigo-Rodriguez [10], Laura M. Parro [11], Cecily Sunday[1,12], Damien Vivet[1], David Mimoun [1], Andrew S. Rivkin [4] & Nancy L. Chabot [4]

Planetary defense efforts rely on estimates of the mechanical properties of asteroids, which are difficult to constrain accurately from Earth. The mechanical properties of asteroid material are also important in the interpretation of the Double Asteroid Redirection Test (DART) impact. Here we perform a detailed morphological analysis of the surface boulders on Dimorphos using images, the primary data set available from the DART mission. We estimate the bulk angle of internal friction of the boulders to be $32.7 \pm 2.5°$ from our measurements of the roundness of the 34 best-resolved boulders ranging in size from 1.67–6.64 m. The elongated nature of the boulders around the DART impact site implies that they were likely formed through impact processing. Finally, we find striking similarities in the morphology of the boulders on Dimorphos with those on other rubble pile asteroids (Itokawa, Ryugu and Bennu). This leads to very similar internal friction angles across the four bodies and suggests that a common formation mechanism has shaped the boulders. Our results provide key inputs for understanding the DART impact and for improving our knowledge about the physical properties, the formation and the evolution of both near-Earth rubble-pile and binary asteroids.

Part of the history and physical properties of an asteroid is recorded in the morphological characteristics of the boulders observable at its surface[1-3]. Geological processes such as impacts[4], thermal processing[5], weathering and erosion[6] leave morphological markers on the boulders at different scales[7]. Past work has shown that the boulder shape can also be linked to specific formation mechanisms[8], and that the angularity of constituent boulders is directly linked to

the mechanical properties (angle of internal friction) of the bulk medium[9-14].

Morphological analysis of sands or crushed sand particles from images is a standard technique for investigating terrestrial sites[15-20]. However, on Earth the morphological analyses are often combined with direct geotechnical testing either in-situ with field studies[6] or in a laboratory setting (e.g., triaxial testing[10,12,13] or direct shear testing[21]).

The combination of techniques allows for better constraints on the characterization and differentiation between different materials. However, despite laboratory mechanical testing of rare asteroid samples[22] and some recent missions that favoured direct surface interactions[23–25], it is still relatively rare for asteroid space missions to make in-situ measurements or physical interactions that can be used to infer the asteroid's geotechnical properties. Images, however, are a common data set[26], as all small body missions are equipped with a camera for remote sensing[27–30]. The ability to infer the physical properties of surface material from remote sensing images is, therefore, a very powerful tool.

The morphology of surface boulders has previously been studied for different planetary bodies including rocks at the Mars Pathfinder landing site[26]. These morphological analyses, in particular the rock roundness, quantifying the sharpness of the corners and edges of a particle[15,16,31], highlight the impact processing and catastrophic flooding the boulders went through, matching predictions from other analyses. Isolated boulder fields on comet 67P/Churyumov-Gerasimenko have also been studied, mainly showing that boulders have similar shapes across 67P's surface[32]. For asteroids, the size distributions[33,34] and the axial ratio of large boulders (>5 m) have been studied, notably, on asteroid (25143) Itokawa[35], asteroid (162173) Ryugu[36] and asteroid (433) Eros[37], but a detailed investigation of their morphology has not yet been performed.

This study focuses on the natural satellite of the S-type asteroid (65803) Didymos: Dimorphos. The secondary of the binary system, was the target of the NASA Double Asteroid Redirection Test (DART) mission[38], where the spacecraft deliberately slammed into Dimorphos to modify the asteroid's trajectory as a planetary defense test[39]. The DART impact produced at least an estimated $\approx 6 \times 10^6$ kg of ejecta[40–42], changed the asteroid's orbital period by 33 min[43], and resulted in a momentum enhancement factor of between 2.2 and 4.9, depending on the (currently unknown) mass of Dimorphos[44]. In the DART impact predictions[45–48], and the subsequent modelling work to interpret the observables[49–51] the internal friction angle of the target asteroid material was one of the key model parameters.

Given that the primary mission objective was to impact Dimorphos, the only instrument onboard the DART spacecraft was the Didymos Reconnaissance and Asteroid Camera for Optical navigation (DRACO)[30]. Here we use a dedicated image-processing pipeline to analyse the last complete image captured by DRACO and compute the detailed morphological characteristics of boulders on the surface of Dimorphos around the DART impact site. The morphological parameters are then used to constrain the internal angle of friction of the asteroid surface material and to investigate the mechanism by which the boulders may have formed. Additionally, we conduct a comparative analysis of the morphology of boulders on Dimorphos using the same image-processing pipeline on carefully selected images of the surfaces of three other rubble-pile asteroids; Itokawa, a 0.32 km diameter large S-type asteroid with estimated 40% porosity[52,53], Ryugu, a 0.90 km diameter large C-type asteroid with estimated 50% porosity[54], and Bennu, a 0.49 km diameter large C-type asteroid with estimated 50% porosity[55] (see Fig. 1). This study aims to provide important constraints on the mechanical properties of rubble pile asteroids, while also providing insights into rubble pile and binary asteroid formation mechanisms.

## Results

### Morphological analysis of boulders at the DART impact site

We conduct an in-depth examination of the last complete image captured by DRACO 2.78 s before the DART impact (Figs. 1a and 2a) to retrieve the boulder dimensions and several dimensionless morphological parameters: roundness, sphericity, circularity, solidity, and two different axial ratios : ellipsoidal ratio and bounding box ratio (see Methods−Determination of morphological parameters). The image

covers an approximate area of 880 $m^2$ on the surface of Dimorphos with a resolution of 0.055 m/pixel[39]. Our analysis centers on 53 selected boulders situated around the impact site of DART for non resolution dependant parameters and 34 boulders for resolution dependant parameters. The process to detect and retrieve the contours of boulders (Fig. 2) was semi-automatized (see Methods−Semi-automatic image segmentation). An automated image analysis pipeline was used for the morphological analysis (see Methods−Determination of morphological parameters). The average values and the standard deviations of all of the computed morphological characteristics are displayed in Table 1. The Dimorphos boulders selected close to the DART impact site have an average equivalent diameter of $2.15 \pm 0.94$ m, are slightly elongated (circle ratio sphericity of $0.61 \pm 0.12$) and sub-rounded in shape (roundness of $0.49 \pm 0.10$), as defined by Wadell[16]. The roundness value of 0.49 should be considered as an upper limit (Methods−Boulder sample selection for smaller scale descriptors). The Dimorphos boulders do not exhibit a large degree of small-scale or large-scale roughness (solidity of $0.95 \pm 0.02$ and circularity of $0.84 \pm 0.09$).

### Implications for Dimorphos' mechanical properties and boulder formation mechanisms

The mechanical properties such as the internal angle of friction are critical parameters with direct implications for the modelling and comprehension of the DART impact event[46,47,49], notably on the behaviour and responses exhibited by the asteroid[46]. We use our calculated roundness characteristics of individual boulders to deduce the regolith's internal friction angle based on an empirical relationship derived in laboratory experiments using particle roundness computed in a similar way[13] (see Methods−Determining the internal friction angle). The average roundness of the boulders at the DART impact site has a mean of $0.49 \pm 0.10$ (Fig. 3). This leads to a mean internal friction angle of at least $32.7 \pm 2.5°$, estimated from a sample of the boulders with diameter >30 pixels at the surface of Dimorphos (See Methods−Boulder sample selection for smaller scale descriptors).

Fragments formed during laboratory impact experiments typically have an average axial ratio ($b/a$) of 0.70-0.74. Moreover, fragments formed by catastrophic disruptions are more oblate ($c/a \approx 0.5$) while fragments formed by impact cratering are flatter ($c/a \approx 0.2$)[8]. Previous work has relied on such experiments to conclude that most boulders of diameters ranging from 5 to 85 m on Itokawa and Ryugu[35–37], have likely been formed by catastrophic disruption based on the observed axial ratios of the surface boulders and boulders on Eros were formed by impact cratering[37]. In these studies, the axial ratios, obtained by fitting ellipses on the boulders observed at Ryugu and Itokawa surfaces, were found to be slightly smaller than in the laboratory experiments (0.68 and 0.63 respectively) whereas boulders at the surface of Eros were less elongated (ratio of 0.72)[37]. However, the observed discrepancy between the ellipsoidal ratios and the laboratory impact ratio is attributed to the inclination of boulders on asteroid surfaces. For Itokawa and Ryugu, where the gravity is lower and then does not force most boulders to lie on their $c$ axes perpendicular to the asteroid surface, which inclination influences the apparent axial ratio. For Eros, the gravity is relatively high, i.e. the $c$ axes of the boulders is most likely to be perpendicular to the surface and thus the apparent axial ratio is closer to the actual axial ratio[37].

The apparent axial ratio has been computed in two ways in this study : with the minor and major axis of a fitted ellipse on a boulder (the ellipsoidal ratio) or with the width and length of the minimum bounding box of the boulder (the bounding box ratio) (see Methods−Determination of morphological parameters). The ellipsoidal ratio tends to be smaller than the bounding box ratio (Table 1, Supplementary Figs. 30 and 31), due to the differences in the axis selection in both methodologies. For Dimorphos, the upper range of the ellipsoid ratio and the bounding box ratio ($0.66 \pm 0.15$ and $0.70 \pm 0.17$, Table 1)

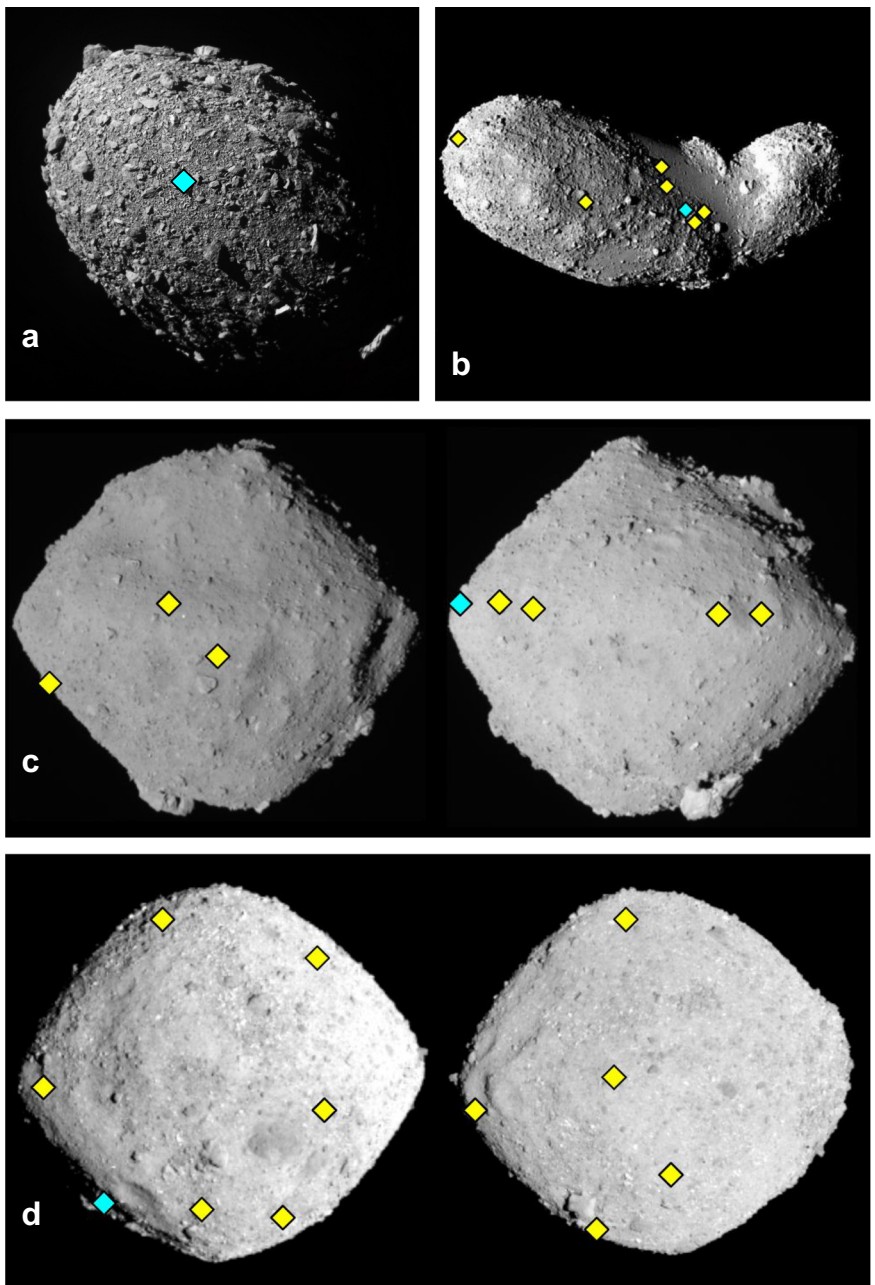

**Fig. 1 | Rubble pile asteroids visited by space missions. a** Dimorphos (208 × 160 × 133 m), the secondary of the (65803) binary system Didymos visited by the NASA DART mission[39], (**b**) Asteroid (25143) Itokawa (607 × 287 × 264 m) visited by the JAXA Hayabusa mission[52], (**c**) Asteroid (162173) Ryugu (1040 × 1020 × 880 m) visited by the JAXA Hayabusa-2 mission[54], (**d**) Asteroid (101955) Bennu (565 × 535 × 508 m) visited by the NASA OSIRIS-REx mission[55]. The diamonds indicates the approximate geographical locations of the images that we used of these asteroids and the blue ones are the images displayed in Fig. 2. Images are not to scale.

also fall within the axial ratio acquired from the laboratory impact fragmentation experiments[8]. Figure 4 shows the histogram and the kernel density estimate (see Methods—Kernel Density Estimate) for the apparent axial ratio. Following the same logic as previous interpretations[35], we can therefore conclude that the boulders on the surface of Dimorphos were likely formed by catastrophic disruption.

## Image selection of rubble pile asteroids surfaces

We selected the most suitable images available for our study of the surfaces of Itokawa, Ryugu and Bennu (Figs. 1 and 2) in order to produce the most reliable morphological analyses possible. These asteroids were selected based on their rubble pile nature[52,54,55] and the high resolution images we have of their surfaces[27,29,56]. The selection of images was based on several criteria: the resolution, the phase angle and, if available, a varied geographical location. (See Methods—Image selection). Based on our image selection criteria, the detailed boulder analysis of these asteroids were determined from 7 images with a total of 277 boulders detected for Itokawa (Supplementary Figs. 1–6), 8 images with a total of 318 boulders detected for Ryugu (Supplementary Figs. 7–13) and 12 images with a total of 955 boulders (Supplementary Figs. 14–24).

The images of Itokawa were taken by the AMICA[27] instrument on board the Hayabusa spacecraft, the images of Ryugu were taken by the ONC-T camera[56] instrument onboard the Hayabusa-2 spacecraft, and the images of Bennu were taken by OCAMS[29] onboard the OSIRIS-REx spacecraft.

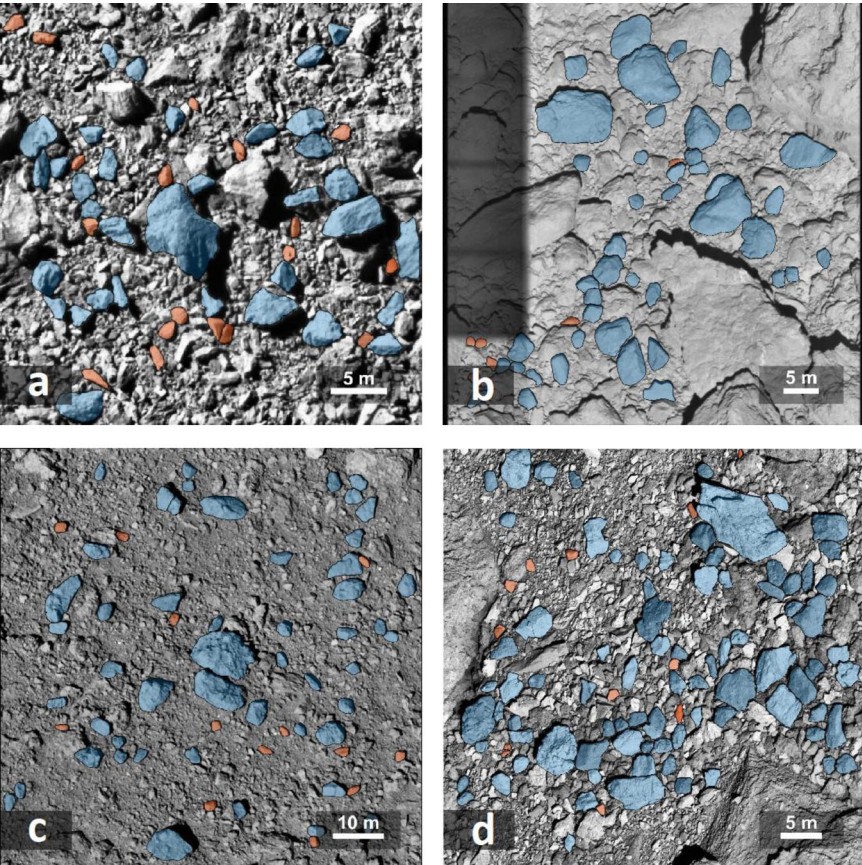

**Fig. 2 | Images of studied asteroids surfaces.** Examples of one of the High-resolution images for each asteroid surfaces studied. The images contrast have been enhanced with a CLAHE filtering for visualisation purposes only[95].
**a** Dimorphos of the (65803) binary system Didymos taken by DRACO[30] (`dar-t_0401930049_43695_02_iof`) (the only image analyzed for this body), (**b**) Asteroid (25143) Itokawa taken by AMICA[27] (`st_2539437177_v`) (1 of the 7 images analyzed), (**c**) Asteroid (162173) Ryugu taken by ONC-T[56] (`hyb2_onc_20180921_041826_tvf`) (1 of 8 images analyzed), (**d**) Asteroid (101955) Bennu taken by OCAMS[29] (`20210407T033629S004_pol_iofL2pan`) (1 of the 12 images analyzed). All other images analyzed can be find in the supplementary information. Boulders selected and analyzed are coloured in blue and red. The red boulders indicate the smaller resolved boulders (<30 px), which haven't been included in the analysis of the resolution dependant morphological parameters.

## Comparative morphological analysis of boulders on rubble pile asteroids

The contours of boulders were detected and retrieved semi-automatically (see Methods−Semi-automatic image segmentation and Fig. 2). The subsequent morphological analysis was performed using the same analysis pipeline as for the image of Dimorphos (see Methods−Determination of morphological parameters). The average values and the standard deviations of calculated morphological parameters are displayed in Table 1 for all studied asteroids.

The boulders analyzed on the surface of the four rubble pile asteroids are all between 0.33 m and 12.15 m. These four rubble pile asteroids exhibit striking similarities in the distributions and average parameters of boulder shapes (Table 1). The mean roundness values span from 0.49 to 0.56, circularity ranges between 0.84 and 0.90 and solidity varies from 0.95 to 0.96. The ellipsoidal axial ratio extends from 0.66 to 0.71 and the bounding box axial ratio extends from 0.70 to 0.75 (Fig. 4). Both of these axial ratios are consistent with previous measurements made for boulders >5 m on Itokawa[37] and Ryugu[36]; 0.62 ± 0.19 and 0.70, respectively.

As for Dimorphos, the roundness values of the surface boulders analysed here can also be used to estimate the angle of internal friction of the surface material of each of these asteroids (Fig. 5) giving values of 32.2 ± 2.5°, 31.6 ± 2.5° and 31.1 ± 2.7° for asteroids Itokawa, Ryugu and Bennu, respectively. These values are consistent with numerical simulations suggesting a friction angle ≲35° for Bennu[57], and the surface slopes on Bennu at the site of the sample collection, which were

≤40°[58,59]. This gives an average value for all of the boulders on the four rubble pile asteroids of 31.8 ± 5.1°. Similarly to the analyses of Dimorphos, we only selected boulders with a diameter >30 pixels (Methods−Boulder sample selection for smaller scale descriptors). We note that these are estimates for the bulk internal friction angle for a material constituted only from the boulders. This provides a reasonable approximation of the bulk internal friction angle for bodies that seem to be dominated by such boulders. However, any finer material present will also influence the bulk material properties, so this value should only be interpreted as the internal friction angle of the boulder-material.

Despite the diverse global morphologies and varying spectral classes (indicative of different lithologies) of these four rubble pile asteroids, the shared resemblance of boulder morphology strongly suggests a common formation mechanism as well as a common response to this mechanism, responsible for shaping the boulders across their surfaces. By comparison with laboratory impact experiments[8], the measured axial ratio for boulders on all four asteroids supports the catastrophic disruption hypothesis[60], as previously suggested for Itokawa and Ryugu[37] and suggested above for Dimorphos.

## Discussion

In the absence of direct measurements of mechanical properties, images can be used to constrain the angle of internal friction of asteroid material through a detailed morphological analysis of

**Table 1 | Mean values of the morphological characteristics of the studied boulders**

| Asteroid | (65803) Dimorphos | | (25143) Itokawa | |
|---|---|---|---|---|
| | All | >30 px | All | >30 px |
| Number of boulders analyzed | 53 | 34 | 277 | 230 |
| Average boulder diameter (m) | 2.15 ± 0.94 | 2.55 ± 0.96 | 2.36 ± 1.71 | 2.60 ± 1.76 |
| Boulder diameter (min/max/median) (m) | 1.07/6.64/1.89 | 1.67/6.64/2.26 | 0.33/10.70/2.12 | 0.66/10.70/2.33 |
| Roundness* | 0.49 ± 0.10 | | 0.54 ± 0.10 | |
| Circularity* | 0.84 ± 0.09 | | 0.90 ± 0.08 | |
| Solidity | 0.95 ± 0.02 | | 0.96 ± 0.02 | |
| Circle ratio Sphericity | 0.61 ± 0.12 | | 0.67 ± 0.12 | |
| Axial ratio: Bounding box | 0.70 ± 0.17 | | 0.74 ± 0.16 | |
| Axial ratio: Ellipsoidal | 0.66 ± 0.15 | | 0.71 ± 0.15 | |
| Asteroid | (162173) Ryugu | | (101955) Bennu | |
| | All | >30 px | All | >30 px |
| Number of boulders analyzed | 318 | 267 | 955 | 703 |
| Average boulder diameter (m) | 3.52 ± 1.67 | 3.77 ± 1.70 | 2.04 ± 0.98 | 2.35 ± 0.96 |
| Boulder diameter (min/max/median) (m) | 1.22/10.39/3.04 | 1.65/10.39/3.29 | 0.51/12.15/1.85 | 1.45/12.15/2.10 |
| Roundness* | 0.56 ± 0.11 | | 0.51 ± 0.10 | |
| Circularity* | 0.90 ± 0.09 | | 0.86 ± 0.09 | |
| Solidity | 0.96 ± 0.03 | | 0.95 ± 0.02 | |
| Circle ratio Sphericity | 0.67 ± 0.11 | | 0.63 ± 0.11 | |
| Axial ratio: Bounding box | 0.75 ± 0.16 | | 0.72 ± 0.16 | |
| Axial ratio: Ellipsoidal | 0.71 ± 0.14 | | 0.68 ± 0.14 | |

Mean and standard deviation values of the morphological characteristics of boulders at the surface of different rubble pile bodies. The boulder diameters reported here are the equivalent diameters.
Morphological characteristics with * are resolution-dependant and are computed only using boulders with diameter >30 px, and these should be considered as higher limits.

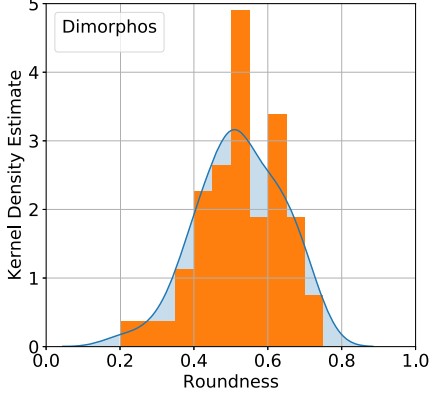

**Fig. 3 | Roundness distributions of the studied boulders.** (Left) Distribution of the roundness of the boulders with a diameter >30 px at the surface of Dimorphos. The orange histogram shows the distribution and the blue shaded region represents the kernel density estimate of the internal friction angle for Dimorphos. (Right) Kernel density estimates of the roundness of surface boulders with a diameter >30 px on Dimorphos (blue with 34 boulders), Itokawa (orange with 230 boulders), Ryugu (green 267 boulders) and Bennu (red with 703 boulders).

the surface boulders. Our analyses show that the boulders on the surfaces of four different small (<1 km) rubble pile asteroids show strikingly similar morphologies suggesting a common boulder formation mechanism as well as a common response to this mechanism. In general, many factors can influence the observed shapes of boulders on planetary surfaces such as their original shapes, their lithology or the nature, violence and duration of the alteration process that the boulders went through[61]. Usually, it is difficult to attribute the observed morphology to one factor such as the formation mechanism. However, in the case of the dry- and atmosphere-less asteroid surface, alteration and erosion processes are likely to be minimal; the material is transported around asteroid surfaces[62,63] but due to the low gravity environment, any motion will occur at low speed, and the small normal force will

reduce friction interactions[64,65]. This decreases the influence of erosion, except for episodic events such as impacts[4]. In the case of impact disruption, it was found that the original target shape does not strongly affect the final fragment shape but are strongly influenced by the kinetic energy of the projectile[8]. Furthermore, although the lithologies of the studied asteroids are different, we still find similar boulder morphologies. Consequently, a shared formation process can explain the similarities in boulder shape amongst the four asteroids. The measured boulder axial ratios support the catastrophic disruption formation hypothesis for small rubble pile asteroids[60], as previously suggested for Itokawa, Ryugu and Eros[37]. Following the same logic as previous work, our results suggest that the boulders on Dimorphos were also formed through catastrophic disruption, later followed by rotation-driven

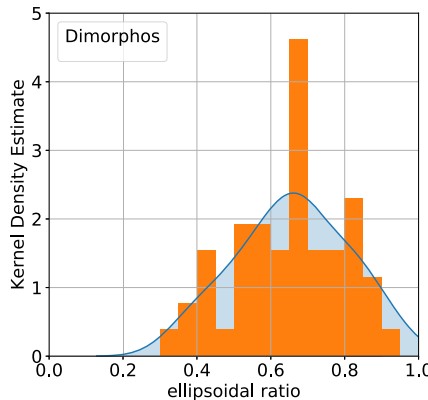
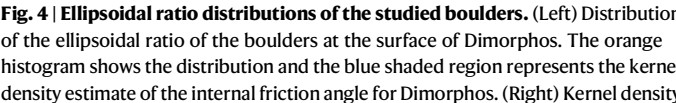
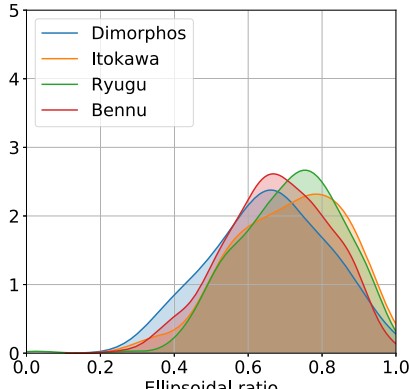

**Fig. 4 | Ellipsoidal ratio distributions of the studied boulders.** (Left) Distribution of the ellipsoidal ratio of the boulders at the surface of Dimorphos. The orange histogram shows the distribution and the blue shaded region represents the kernel density estimate of the internal friction angle for Dimorphos. (Right) Kernel density estimates of the ellipsoidal ratio of surface boulders on Dimorphos (blue with 53 boulders), Itokawa (orange with 277 boulers), Ryugu (green with 318 boulders) and Bennu (red with 955 boulders).

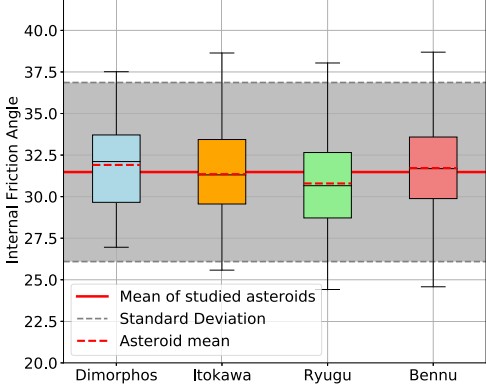

**Fig. 5 | Internal friction angle as derived from the roundness of the studied boulders.** Boxplots of the internal friction angles values from the boulders >30 pixels analyzed on the images of Dimorphos 34 boulders, Itokawa (230 boulders), Ryugu (267 boulders) and Bennu (703 boulders), see Fig. 2. The red dotted line is the mean internal friction angle of boulders on each asteroid, the black line is the median, the limit of the boxes are the lower and upper quartiles of the distribution and the limits outside the boxes are the lower and upper extremes. The red line in the background is the average of the 4 asteroids and the shaded area lies between the propagated standard deviation between all 4 asteroids.

mass transfer between the two satellites in the binary system. This implies either that catastrophic disruption was involved in the binary asteroid formation, or more likely that the process forming the secondary did not significantly modify the boulder morphology, since we are still witnessing morphological markers indicating catastrophic disruption. Also note that Dimorphos have the smallest value of mean axial ratios probably because the image was taken with a higher phase angle than the other images of asteroids, which can influence the shape distribution[35].

Other processes could shape boulders at the surface of asteroids such as thermal fatigue[66,67]. Indeed, cracks that may have been formed by such a process have been identified at the surface of Dimorphos and it is predicted that 10-100 Myr are needed to break the boulders[68]. However, Dimorphos' surface is relatively young (≤0.3 Myr)[69]. It is, therefore, unlikely that thermal processing has had sufficient time to reshape boulders after the formation of the binary asteroid[70].

From our measurements of the angularity of individual boulders, we constrain what should be considered a lower limit (Methods– Boulder sample selection for smaller scale descriptors), the mean bulk

angle of internal friction of the surface boulders of these rubble pile asteroids to be $31.8 \pm 5.1°$. The bulk angle of internal friction of the ensemble of boulders at the DART impact site is found to be $32.7 \pm 2.5°$, indicating that Dimorphos boulders are typical rubble pile boulders, and that the angle of internal friction of a rubble pile is independent of spectral type and, by extension, composition. However, we note that certain characteristics such as porosity, are not accessible through a morphological analysis.

In addition to providing information about the behaviour of asteroid surfaces, our findings are useful for the continued interpretation of the DART impact, for planetary defense efforts that rely on estimates of asteroid mechanical properties in order to estimate the potential damage an asteroid may cause if it were to impact the Earth[71–74], and to evaluate the effectiveness of key asteroid deflection techniques[75]. The insights about the likely boulder formation mechanism will be useful for improving our knowledge about the formation and evolution of both rubble pile asteroids and binary asteroids.

In the future, it would be interesting to include more images of Dimorphos' surfaces in these analyses. The European Space Agency Hera mission[76] will provide global images of the surfaces of both Dimorphos and Didymos allowing more detailed analyses to be made of the DART target, and allowing for a comparison of the boulder morphology on the surfaces of the primary and secondary asteroids.

## Methods
### Image selection
The images of Itokawa, Ryugu and Bennu have been selected with the following criteria : the pixel resolution must be under 0.1 m/px; if available, the phase angle should be under 20° to minimize boulder shadowing effects; and finally if available, we tried to select images from varied locations on the surface of the asteroids. Based on the pixel resolution and phase angle criteria, 7 images of Itokawa were selected. These images are located at 2 different areas on the asteroid with a total of 277 boulders identified semi-automatically. On Ryugu, 8 images were selected based on the distance of the spacecraft with the asteroid (i.e. resolution) but the phase angles were not available. However, each image has been randomly selected at 8 different locations in order to avoid overlapping images, where the spacecraft was close enough to the asteroid to retrieve well-resolved images of its surface for a total of 318 boulders semi-automatically identified. Finally, 12 images were selected at 12 different locations at the surface of Bennu. The images retrieved by the OSIRIS-REx missions offered an

almost global coverage of very high resolution images. For this reason, we chose 12 separate locations on Bennu's surface and selected an image around the location following the criteria cited above for a total of 955 boulders semi-automatically identified. The detailed data of each images are detailed in Supplementary Table 1.

## Semi-automatic image segmentation

The images selected for the study were segmented in a two-step process using a Python package, `segmenteverygrain`[77], that enables to detect grain-like objects in images. This `segmenteverygrain` package uses a machine learning model to obtain precise and accurate outlines of the grains. It also possess functionalities to let the user delete and merge objects, and add grains that were not segmented automatically at first, by clicking inside the grain outline.

The first step of the process segments the raw images selected as input (Supplementary Fig. 25a) and the boulders outlines can be retrieved. However, depending on the images, visible errors reside in the detection of the boulders outline. To improve the quality of the boulder morphology analysis, a second step is necessary: manual confirmation by the user of the detected boulders and adding any boulders that may have not been selected automatically in the first step (Supplementary Fig. 25b, outlines are always retrieved automatically). The selected boulders had to verify one main criterion: they have to not be cut-off from the image edges, or buried under some regolith material that may be hiding a significant part of the boulder, or either hiding in the shadow (from the cast or self shadows). This criteria is checked manually, and, therefore, contains some component of subjectivity. All boulders of the images were checked, and when a doubt existed, the object was discarded to minimize any subjectivity from the human interpretation.

This semi-automatic method enables to process more images in a smaller amount of time than if the images were segmented completely manually and therefore allows to analyse more areas on asteroids. Even if more boulders could be identified manually on a single image, this method decreases the biases from human interpretation in the boulder outline definition.

## Determination of morphological parameters

Most morphological parameters studied in this paper are dimensionless quantities used in image analysis to describe the shapes of particles regardless of their sizes. The pipeline used for this has been developed in MATLAB. From the particle contour coordinates, it automatically computes the morphological parameters. The definitions and method of calculation of the morphological parameters provided in Table 1 are provided below.

The metric used as the size parameter is the equivalent diameter[16], $D_c$, is the diameter of the circle with the same projected area as the particle (Supplementary Fig. 26):

$$D_c = 2*\sqrt{A/\pi} \tag{1}$$

with $A$ being the projected area of the particle ("area" property of the Image Processing MATLAB Toolbox *regionprops* function).

The circularity[78], $C$, (or compactness) of a particle represents how closely the polygon shape of that particle matches a circle of same area (Supplementary Fig. 26):

$$C = \frac{4\pi A}{P_S^2} \tag{2}$$

where $P_S$ is the perimeter of the particle, computed by counting the pixels defining the boulder edge ("perimeter" property of the Image Processing MATLAB Toolobox *regionprops* function). The circularity is a measure of the large-scale roughness of a particle. A circularity of 1 indicates that the particle is circular.

The solidity or convexity ratio[79] represents how closely does the shape of a particle match its own convex hull (Supplementary Fig. 27):

$$\text{Solidity} = \frac{A}{H} \tag{3}$$

with $H$ the area of the convex hull of the particle ("ConvexArea" property of the Image Processing MATLAB Toolobox *regionprops* function). It is a measure of the small-scale roughness of the boulders. A solidity of 1 indicates that the particle is its own convex hull.

The axial ratios of a 3D shape are the ratios between the dimensions in the three orthogonal plane of the particle, often referred as $a$, $b$, and $c$ ($a \geq b \geq c$). When analysing 2D images, the particle axial ratio can only be studied with its two apparent axis. The apparent axial ratio can be computed two different ways:

- Ellipsoidal ratio (Supplementary Fig. 28): the axial ratio is computed with the minor and major axis of the fitted ellipse of the particle.
- Bounding box ratio (Supplementary Fig. 29): the axial ratio is computed with the width and length of the minimal bounding box of the particle.

$$\text{Ellipsoidal ratio} = \frac{b}{a} \tag{4}$$

$$\text{Bounding box ratio} = \frac{d_2}{d_1} \tag{5}$$

with $a$ and $b$ being the major and minor axis of the fitted ellipse of the particle, respectively ("MajorAxisLength" and "MinorAxisLength" properties of the *regionprops* function), and $d_1$ and $d_2$, the minimal bounding box length and width, respectively. The bounding box is a rectangle which include all coordinates of the particle outline, and which minimize the area of the rectangle. This one is found using a preexisting MATLAB function[80].

Both the ellipsoidal and bounding box ratio can be used to obtain the apparent axial ratio of a particle. An ellipsoidal or a bounding box ratio of 1 indicates that the studied particle is equidimensional. The ellipsoidal ratio has been used in previous studies[36,37,70] to measure the axial ratio of asteroid boulders. These methods, are similar, but do differ from each other, as seen in the results when using both methodologies to compute the axial ratios (Supplementary Figs. 30 and 31).

Sphericity, $S$, is a three-dimensional parameter that can also be studied in two dimensions. In three dimension, it is the relationship of the shape of a particle to a sphere and the relationship of the shape of a particle to a circle in two dimensions. It can also be referred as the circle ratio sphericity[81] (Supplementary Fig. 32):

$$S = \frac{D_{ins}}{D_{cir}} \tag{6}$$

with $D_{cir}$ the diameter of the minimum circumscribing circle, and $D_{ins}$ the diameter of the largest inscribing circle. A sphericity equal to 1 indicates that the particle is a circle, with the minimum circumscribing circle being equal to the maximum inscribing circle.

A previously defined methodology[82] is used to find the minimum circumscribed circle of a particle (red circle in Supplementary Fig. 32). In this method, the concave points around the contour are used to define a first circle of a diameter corresponding to the largest distance between two points. If all points of the contour are inside the circle, the minimum circumscribed circle has been found. Otherwise the procedure is repeated by including the outside point the furthest away from the previous circle to fit a new circle from 3 points.

The maximum inscribed circle (blue circle in Supplementary Fig. 32) is found by computing the Euclidian distance transform of the particle[83] with the *bwdist* function in the Image Processing MATLAB ToolBox. The distance transform computes the minimum distance to the contour from each pixel inside the particle. The maximum value of the distance transform map corresponds to the most enclosed point of the particle that is the furthest away from the contours. The maximum inscribed circle has a radius equal to the maximum value of the distance transform.

Roundness, $R$, indicates the relative curvature of the projected shape of the particle. It quantifies the sharpness of its corners and edges[15,16,31]. The definition of roundness[15] is the ratio of the average radius of curvature of the corners of the particle to the radius of the maximum inscribed circle (Supplementary Fig. 33):

$$R = \frac{\sum_{i=1}^{N} r_i}{N \times r_{ins}} \tag{7}$$

with $N$ being the number of corners, $r_i$ the radius of each corner circle and $r_{ins}$ the radius of the largest inscribing circle. Higher roundness indicates the particle is more round and lower roundness indicates the particle is more angular.

In order to find the corners of the particle and their associated radius $r_i$, the discrete curvature around the smoothed contour is computed. The contour is smoothed by 5% with the LOESS (LOcally Estimated Scatterplot Smoothing)[84] method as a means to remove the angular/square shape contour from the pixels. The curvature $\gamma$ of a 2D closed line is the second derivative of the contour coordinates. A curvature equal to 0 corresponds to a straight line. A negative ($\gamma < 0$) and positive ($\gamma > 0$) curvature are concave and convex points, respectively. For one given point of curvature $\gamma_p$, the associated radius $r_p$ is its inverse, as $r_p = \frac{1}{\gamma_p}$.

In Wadell's definition[15], the corner circle's radius cannot be larger than the radius of the maximum inscribed circle ($r_i <= r_{ins}$). Therefore, a corner is defined by being successive concave points of a curvature higher or equal to $1/r_{ins}$. For each corner, the local maximum of the curvature is identified, and corresponds to the radius of the biggest circle that can fit in the corner.

The roundness of two additional samples was measured in order to emphasize and demonstrate the roundness differences in granular samples. For the angular sample, an image of broken glass was analysed (Supplementary Fig. 34a), as for the rounded sample, an image of pebbles found on the surface of Mars was used for the analysis (Supplementray Fig. 34b). These images and samples were selected as extreme cases for the roundness computation of granular samples. The broken glass and pebbles on Mars has an average roundness and standard deviation of $0.26 \pm 0.10$ and $0.69 \pm 0.07$, respectively. The normal distribution of the roundness for these two samples are compared with the roundness distribution of the boulders of Dimorphos (Supplementary Fig. 35). The distributions of the three different samples enables to demonstrate the differences in range in the roundness computation for different granular materials.

### Boulder sample selection for smaller scale descriptors
The resolution of the image, or more precisely, the size in pixels of a particle, may affect the results of the morphological descriptors, especially for the smaller scale descriptors such as the roundness[82], and circularity[85]. All larger scale descriptors (sphericity, solidity, and axial ratios) are not resolution-dependant. Thus, in order to avoid computational errors and increase the reliability of our results we selected boulders with diameters >30 pixels to compute the roundness and circularity. The 30 pixels threshold size was defined for our analysis, in order to have a large enough sample size for each body studied (>30 boulders[86,87]) to have reliable statistics. Dimorphos was

the limiting case, where only one image was analysed: to have a sample size of at least 30 boulders to compute the smaller scale descriptors, all boulders with a resolution above 30 pixels were included. For the circularity, the error lies in the calculation of the perimeter; with too few pixels defining the outline of the particle, the perimeter will be underestimated. This behaviour can result in a circularity exceeding the value of 1, as the circularity is the ratio between the perimeter of the particle and the perimeter of a circle with the same area. As for the roundness, if the particle is made of a few too pixels, then its corners curvature are underestimated, and results in higher roundness values computed. For the two smaller scale descriptors, roundness and circularity, the values were computed with all boulders with a diameter >30 pixels. The roundness values presented here should be considered as a higher limit due to the low resolution of the sample. As for the angle of friction derived from the roundness, this should be considered as a lower limit.

### Determining the internal friction angle
The empirical relationship between the bulk internal friction angle, $\phi$, and average roundness of the constituent particles[13], $R$, is:

$$\phi = 25.02 \times (1 - R) + 20 \tag{8}$$

The relationship between the roundness distribution and the resulting angle of internal friction distribution is displayed as bivariate kernel density (see Methods−Kernel Density Estimate) in Supplementary Fig. 36.

This relationship has been obtained in laboratory experiments studying several types of sands, crushed sands and glass beads. The roundness of the particles was computed from binarized 2D images with a MATLAB code[82]. The internal friction angles ($\phi$) of these different soils were measured using a triaxial shear test. The experiment measures under which shear stress a material fails for a given normal stress. The internal angle of friction (along with the material cohesion) is then determined from the Mohr-Coulomb law[88]. Given that these parameters are intrinsic to the material, the results are not gravity dependant, unlike surface slopes or the angle of repose (often used as a proxy for the internal angle of friction), which can vary under different gravitational conditions[89].

There are also three other equations linking roundness and the bulk internal friction angle following the same experimental approach, but the method for calculating roundness is different from that of this paper: one is measuring roundness by comparing a sample of grains with the Sloss & Krumbein roundness chart[10,18], another one is using the combined roundness[11], taking in account mixed materials[12] and the last one taking in account the coefficient of uniformity of the material[14].

### Kernel Density Estimate
Kernel density estimation[90] or the Parzen's window[91] is an approach to estimate the underlying probability function of a data set[92]. In our case, it smooths the histograms of our data to compare them more effectively (For example, see Fig. 4). This smoothing is the reason why values can exceed 1 on Fig. 4 even if no value exceeds 1 in the dataset. The bivariate density estimation[93] shows the correlation between kernel density estimates of two variables from the same data set (See Supplementary Fig. 36). It is computed using the seaborn Python library[94] with the *kdeplot* and *jointplot* functions.

## Data availability
The DART DRACO, the Hayabusa AMICA, the Hayabusa 2 ONC and the OSIRIS-Rex OCAMS data are publicly available in the Planetary Data System : DRACO AMICA ONC OCAMS Source data generated in this study have been deposited in the Zenodo database: https://doi.org/10.5281/zenodo.10848982 Source data are provided with this paper.

**Article**

## Code availability

The `segmenteverygrain` code used to semi-automatically detect the boulders contours is available on GitHub: https://github.com/zsylvester/segmenteverygrain. The code used to retrieve the morphological parameters of the boulders is available on demand to the corresponding author.

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

## Acknowledgements

N.M., C.R. and P.M. acknowledge funding support from the European Commission's Horizon 2020 research and innovation programme under grant agreement No 870377 (NEO-MAPP project). N.M. and P.M. acknowledge support from the Centre National d'Etudes Spatiales (CNES), focused on the Hera space mission. C.R. acknowledges PhD funding from CNES and ISAE-SUPAERO. A.D. acknowledges PhD funding from University of Toulouse III. This work was supported by the DART mission, NASA Contract No. 80MSFC20D0004. A.L. and M.P. acknowledges support by the Italian Space Agency (ASI) within the LICIACube project (ASI-INAF agreement n. 2019-31-HH.0) and HERA project (ASI-INAF agreement n. 2022-8-HH.0). S.D.R. acknowledges support by the Swiss National Science Foundation (project number 200021_207359). P. M. acknowledges support from The University of Tokyo. A.E.S. is supported by the CNES postdoctoral fellowship. L.M.P. contribution was supported by the Margarita Salas postdoctoral grant funded by the Spanish Ministry of Universities—NextGenerationEU and CIAPOS/2022/066 postdoctoral grant (European Social Fund). JMT-R acknowledges support from the Spanish project PID2021-128062NB-I00 funded by MCIN/AEI.

## Author contributions

C.R. led the data analysis and writing of the paper. A.D. helped to develop the image processing pipeline, performed the semi-automatized image segmentation and contributed to the data analysis and paper writing. N.M. was the supervising author for this study, made significant contributions to the writing of the paper and guided the analysis and interpretations. J.-B.V. contributed initial binarized images of the boulders at the surface of Itokawa and Ryugu that were used at the beginning of this study. R.T.D., O.S.B. and C.M.E. helped develop foundational data products, including DRACO image calibration, and provided significant comments on the paper. M.H., S.D.R., M.P. and A.L. helped give current understanding of geological properties in rubble pile asteroids and interpretations of consistent friction angles over different asteroids found in this study and the review of the work. G.C. and D.V. helped developing the morphological analysis pipeline. A.E.S. helped with the data processing and the review of the work. P.M. helped with the interpretation of the results and provided significant comments on the paper. C.S., L.M.P., E.R.J., D.M. and J.M.T.-R. helped to improve the quality of the manuscript. N.L.C. and A.S.R. helped with their management role in the DART space mission.

## Competing interests

The authors declare no competing interests.

## Additional information

[1]Institut Supérieur de l'Aéronautique et de l'Espace (ISAE-SUPAERO), Université de Toulouse, Toulouse, France. [2]DLR Institute of Planetary Research, Berlin, Germany. [3]INAF-Astronomical Observatory of Padova, Padova, Italy. [4]Johns Hopkins University Applied Physics Laboratory, Laurel, MD, USA. [5]University of Bern, Bern, Switzerland. [6]Côte d'Azur University, Côte d'Azur Observatory, CNRS, Lagrange Laboratory, Nice, France. [7]The University of Tokyo, Department of Systems Innovation, School of Engineering, Tokyo, Japan. [8]Georgia Institute of Technology, Atlanta, GA 30332, USA. [9]Smithonian National Air and Space Museum, Washington, DC, USA. [10]Institute of Space Sciences (CSIC-IEEC), Campus UAB, Carrer Can Magrans s/n, Cerdanyola del Valles, Barcelona, Catalonia, Spain. [11]IUFACyT, Alicante University, San Vicente del Raspeig, 03080 Alicante, Spain. [12]University of Maryland, College Park, MD, USA. ✉ e-mail: colas.robin@isae-supaero.fr

