## [Peer Review File · Nature Communications]

Mechanical properties of rubble pile asteroids (Dimorphos, Itokawa, Ryugu, and Bennu) through surface boulder morphological analysisREVIEWER COMMENTS

Reviewer #1 (Remarks to the Author):

General comments

The authors investigated the morphology of the boulders on Dimorphos, Itokawa, Ryugu, and Bennu, using various morphological parameters. The methods used by the authors are well described and may become a cited example of how to undertake such tasks. That is, the methods are useful for understanding quantitatively the morphology of the boulders on small asteroids. They estimated the internal friction angles on these asteroids to understand the mechanical properties of rubble pile asteroids. As a result, it was found the similarities in the morphology of the boulders on the four asteroids. The authors suggested that the boulders on the four asteroids were primarily produced by impact processing.

However, I don't think that the present set of data are conclusive enough to draw firm structural conclusions although the conclusion may be correct. I think substantial revisions are needed. There are four serious issues that the authors should address prior to acceptance of the article.

(1) The authors analyzed the images (Figs. 2c and 2d) derived from MASCOT for Ryugu and OCAMS for Bennu, respectively. Since these images are the data from oblique imaging of the asteroid's surface, there are significant differences in the resolution between the foreground and the background boulders. Consequently, the authors likely measured the distorted shape of the boulder. To investigate the exact shape of the boulders, it is advisable to use images taken from the front (i.e., where the emission angle is close to 0 degrees) whenever possible. In this regard, the images presented in Figs. 2c and 2d are not be the suitable choice. Therefore, the authors should consider reanalyzing the data using different images for Fig. 2c and 2d.

(2) The authors manually measured the contour of the boulder excluding its shadow. I understand that it is difficult to measure the contour of the boulder including its shadow. However, part of the real contour of the boulder disappears under the shadow. In my experience, when the axial lengths of the boulders are determined excluding the shadows, the width-to-length ratio becomes small compared to that including the shadow. This may be the reason the width-to-length ratios (Table 1) measured by the authors is smaller than the previously measured ratio of ~ 0.7 . It is also possible that the slight difference in the width-to-length ratios in Table 1 is simply a difference in shadow length (i.e., the width-to-length ratios for Figs. 2a and 2d, which have longer shadows, are smaller than those for Figs. 2b and 2c, which have shorter shadows). Essentially, when comparing the boulder shapes on the four asteroids, the author should use the image data that have a similar angle of solar incidence (i.e., the similar shadow length). If this is difficult, the influence of the shadow on the boulder shape should be explained in the text.

(3) In the text (L90-92), the authors described that the highest resolution images of the surfaces of Itokawa, Ryugu, and Bennu were used for their analysis. I don't agree with this. At least, in the case of Itokawa and Ryugu, there must be other images with higher resolution than the images of Figs. 2b and 2c. Moreover, I doubt that the images selected by the authors (Figs. 2b, 2c and 2d) are the representative of the three asteroid surfaces. Please discuss the bias of the observation points in detail.

(4) The authors estimated the internal friction angle based on an empirical equation (eq. (11)) obtained

from laboratory experiments on the Earth using the average roundness of the particles. However, since the gravity on small asteroids is very different from that on the Earth, I doubt that the empirical equation is applicable to low-gravity asteroids. The authors should explicitly discuss the relationship between the empirical equation and the gravity.

Specific comments

Fig. 1, "rubble pile asteroids..."
Typo? "Rubble pile asteroids..."

Fig. 1c, d, "The yellow square indicates the geographical location of the highest resolution image that we used acquired of the surface of each of these asteroids."
The yellow squares do not correspond to the images shown in Fig. 2. Please specify the location.

Fig. 2,
Why is there no image number listed?

L67, "..., are elongated (width-to-length ratio of 0.60 ± 0.16) and ..."
As described before, it is possible that the width-to-length ratio of 0.60 is small due to the definition of the boulder contour excluding its shadow. Please discuss the relationship between the boulder contour and the shadow.

L76,
It is strange to see that Fig. 4 is placed before Fig. 3.

L79-81,
I think the authors have misunderstood these papers (35,36,37). For example, Michikami et al. (2022) claimed that, almost all of the boulders on Eros are the products of impact cratering, while most of the boulders on Ryugu and Itokawa are primarily products of the catastrophic disruption (section 4.2 of their paper).

Fig. 3,
The authors showed the kernel density estimation in Fig. 3. The value of the width-to-length ratio must be between 0 and 1. However, the kernel density estimation includes the value greater than 1. In this regard, it is strange to use the kernel density estimation in principle. Anyway, please clearly explain the value greater than 1. In addition, please clearly define "the bin of the width-to-length ratio".

L106-108,
Why didn't the author investigate the boulders on the close-up images of Eros? If the authors examine the morphology of the boulders on Eros, more useful information might be obtained in combination with the data of Fig. 4.

L127, "By comparison with laboratory impact experiments (35;36;37), ..."
These references are not the papers in terms of laboratory impact experiments. Please make sure that references are properly cited in the text.

L218,
Is the denominator of the eq. (10) multiplied N by r_{ins} ? If so, please show it explicitly.

Reviewer #2 (Remarks to the Author):

This is a neat study of rubble pile asteroids and their mechanical properties via image analyses of surface boulders. The manuscript is well written. Here, I list a few comments that I trust would elevate the study in terms of its readability and reproducibility.

1. Describe how is angle of friction derived
2. You describe a pipeline for high res image analysis. Will you make this code publicly available as part of data repository? It is hard to judge its product without the code.
3. Some boulders were hand picked and checked against the average properties. This is fine, but there is no discussion how are boulders picked from the images overall in the complete dataset. Could you discuss the limitations in using the selected high resolution images? Any other bias coming from using humans instead of ML methods, etc. Have you thought about applying pattern

recognition methods (this may be beyond the scope, but still worth mentioning)

4. Also, how are high resolution images selected? Can that subset of images be compiled in the supplementary materials? If not that, then provide a list of image IDs that can then be pulled from the PDS. This is to insure reproducibility of your work.

5. It is not very clear (at least to me from the first read) what are elements compared with previous studies and what is done in this work beyond the image analyses from DART. Did the team process the other 3 asteroids image sets, or was that data used from elsewhere? I would think that this is all your work here, but not much of it was clarified in the introduction/abstract, hence my confusion.

Hope this helps.

Reviewer #3 (Remarks to the Author):

This study reports the estimated internal friction angle of various asteroids. The method of estimating internal friction angle is based on an empirical relation between internal friction angle and the average roughness of particles. Various morphological features of the best 30 boulders picked up from asteroids Dimorphos, Itokawa, Ryugu, and Bennu were measured and analyzed. As a consequence, this study reveals that the boulder shapes are similar among these bodies. Particularly, the representative value of the internal friction angle (for rubble-pile asteroids) was estimated as 34 ± 6 degrees. This value seems to be almost independent of details of asteroids. This result is informative for those who want to perform numerical simulations of the asteroidal internal structure. In general, the internal properties of asteroids are difficult to estimate. This study provides a possible answer to that problem. Besides, the aspect ratio of measured boulders is also roughly independent of the details of asteroids. The distribution of the aspect ratio is almost universal. The average value suggests the origin of boulders is probably catastrophic impact. The feature particular for asteroid Itokawa is relatively narrow distribution. According to the discussion of this study, the reason for this is the migration-induced segregation of boulders. The analyses seem to be carried out very carefully and the results obtained are convincing. However, at the same time, the reported results are not very surprising. Therefore, I think this paper deserves publication in more specialized journals. While this manuscript is well written, I have some (relatively minor) comments as listed below.

1. This paper basically insists the difference among the focussed asteroids is almost negligible. The estimated average value of internal friction is close to the usual granular one. In other words, the result is natural and researchers have assumed more or less similar internal friction value. If the difference between different asteroids can be quantitatively revealed by this analysis, it must be very interesting. However, the current measurement uncertainty prevents us from such precise discussion. In this sense, the result is not very surprising. More precise measurements are expected as future problems. The required measurement precision to find the difference among the asteroids should be discussed more quantitatively (based on the current measurement results), if possible.

2. From asteroidal surface terrain, the distribution of local slopes relative to the local horizontal plane can be measured. From this method, a typical angle (of repose) of Ryugu was estimated as 31 degrees (Watanabe et al. Science 364, 268 (2019), Supplementary Material). Although the internal friction angle and angle of repose are different quantities, they usually show similar values, and this is the case in this study, too. Some discussion about the relation between some friction angles should be discussed.

3. Actually, the form Eq.(11) simply interpolates two extreme values 20 degrees and 45 degrees by using a parameter R . The plot of $\phi(R)$ and 4 values corresponding to each asteroid should be presented. Such a plot visually indicates how close the roundness of asteroids, is directly.

4. In Fig. 11, the distribution of Ryugu (with 30 best-resolved) shows a bimodal tendency. This means that the typical boulders are not well resolved. The reason why this study can neglect this tendency should be clearly mentioned.

05/03/2024

Dear reviewers,

We are delighted that you liked this paper and we thank you for your very constructive suggestions to improve the quality of the analyses. We have made significant modifications that include analysing many more images of the surfaces of Bennu, Itokawa and Ryugu. We have also implemented a semi-automatised image segmentation in order to reduce human bias in the selection of boulder contours. Note that we also removed figure 5 of the submitted version of the paper where Itokawa was displaying a tighter distribution in small- and large-scale roughness. It was probably because we were only analysing 1 image from Itokawa in the first version of the paper, because we do not see a particular difference between the distributions of the 4 asteroids now. The detailed responses to all of the reviewer comments are indicated in blue below.

We note also that we have made one change to the author order: A. Duchene has made very significant contributions to the updated paper (including performing the image segmentation and analysis) and we request to move her from 3rd to 2nd author.

Sincerely,

C. Robin and coauthors

REVIEWER COMMENTS

Reviewer #1 (Remarks to the Author):

General comments

The authors investigated the morphology of the boulders on Dimorphos, Itokawa, Ryugu, and Bennu, using various morphological parameters. The methods used by the authors are well described and may become a cited example of how to undertake such tasks. That is, the methods are useful for understanding quantitatively the morphology of the boulders on small asteroids. They estimated the internal friction angles on these asteroids to understand the mechanical properties of rubble pile asteroids. As a result, it was found the similarities in the morphology of the boulders on the four asteroids. The authors suggested that the boulders on the four asteroids were primarily produced by impact processing.

However, I don't think that the present set of data are conclusive enough to draw firm

structural conclusions although the conclusion may be correct. I think substantial revisions are needed. There are four serious issues that the authors should address prior to acceptance of the article.

(1) The authors analysed the images (Figs. 2c and 2d) derived from MASCOT for Ryugu and OCAMS for Bennu, respectively. Since these images are the data from oblique imaging of the asteroid's surface, there are significant differences in the resolution between the foreground and the background boulders. Consequently, the authors likely measured the distorted shape of the boulder. To investigate the exact shape of the boulders, it is advisable to use images taken from the front (i.e., where the emission angle is close to 0 degrees) whenever possible. In this regard, the images presented in Figs. 2c and 2d are not be the suitable choice. Therefore, the authors should consider reanalysing the data using different images for Fig. 2c and 2d.

For Itokawa, Ryugu, and Bennu, several images have now been carefully selected and analysed instead of just one (7, 8 and 12 images, respectively). The images have been selected with a resolution smaller than 0.1 m/px, a phase angle smaller than 20° (when available), and at different locations (if available).

Lines 97- 108 (in Results – Image selection of rubble pile asteroids surfaces)

Lines 188 – 201 (in Methods – Image selection)

The only images with an emission angle close to 0 ($e < 5^\circ$) were available on Bennu and taken during the departure flyby (2 images of this flyby has been analysed)

However, emission angle as well as phase angle were also found to have little to no incidence on the shapes of boulders whether for the difference between foreground and background for roundness and sphericity (Yingst & al., 2007) or even for the distorted shapes for circularity and solidity (Cambianica & al., 2019).

R. Aileen Yingst, A. F. C. Haldemann, Kimberly L. Biedermann, and Aimee M. Monhead. Quantitative morphology of rocks at the Mars Pathfinder landing site. *Journal of Geophysical Research: Planets*, 112(E6), 2007. _eprint: <https://onlinelibrary.wiley.com/doi/pdf/10.1029/2005JE002582>

P. Cambianica, G. Cremonese, G. Naletto, A. Lucchetti, M. Pajola, L. Penasa, E. Simioni, M. Massironi, S. Ferrari, D. Bodewits, F. La Forgia, H. Sierks, P. L. Lamy, R. Rodrigo, D. Koschny, B. Davidsson, M. A. Barucci, J. L. Bertaux, I. Bertini, V. Da Deppo, S. Debei, M. De Cecco, J. Deller, S. Fornasier, M. Fulle, P. J. Gutiérrez, C. Güttler, W. H. Ip, H. U. Keller, L. M. Lara, M. Lazzarin, Z. Y. Lin, J. J. López-Moreno, F. Marzari, S. Mottola, X. Shi, F. Scholten, I. Toth, C. Tubiana, and J. B. Vincent. Quantitative analysis of isolated boulder fields on comet 67P/Churyumov-Gerasimenko. *Astronomy and Astrophysics*, 630: A15, October 2019. ADS Bibcode: 2019A&A...630A..15C

(2) The authors manually measured the contour of the boulder excluding its shadow. I understand that it is difficult to measure the contour of the boulder including its

shadow. However, part of the real contour of the boulder disappears under the shadow. In my experience, when the axial lengths of the boulders are determined excluding the shadows, the width-to-length ratio becomes small compared to that including the shadow. This may be the reason the width-to-length ratios (Table 1) measured by the authors is smaller than the previously measured ratio of ~ 0.7 . It is also possible that the slight difference in the width-to-length ratios in Table 1 is simply a difference in shadow length (i.e., the width-to-length ratios for Figs. 2a and 2d, which have longer shadows, are smaller than those for Figs. 2b and 2c, which have shorter shadows). Essentially, when comparing the boulder shapes on the four asteroids, the author should use the image data that have a similar angle of solar incidence (i.e., the similar shadow length). If this is difficult, the influence of the shadow on the boulder shape should be explained in the text.

After redoing the analyses, the apparent aspect ratio is now ~ 0.7 . We distinguished two methodologies to compute the aspect ratio: the ellipsoidal ratio (ratio of the minor and major axis of the fitted ellipse of the boulder), and the bounding box ratio (ratio of the width and length of the minimal bounding box of the boulder). The difference between the previous and present analysis is in the boulder selection. The process has been semi-automatized, and the criteria to select a boulder more restrictive: when a doubt existed in the boulder outline (due to shadows, or other effect), it was not selected to be part of the analysis. This is now described in Line 202 – 219 (Methods – Semi-automatic image segmentation).

The phase angle is now also one of the criteria in the selection of the images, in order to reduce the effect of shadowing. See line 189 (in Methods – image selection), and Table 2

We also note that the phase angle on the Dimorphos penultimate image is fairly high but with the strict criteria on the boulder selection especially regarding shadows, we should minimize the effect. However, we can see that the mean value for the ellipsoidal ratio on Dimorphos is slightly smaller than on other asteroid. This could be the reason and it has been added in the text.

Line 117-119 (in Results - Comparative morphological analysis of boulders on rubble pile asteroids)

(3) In the text (L90-92), the authors described that the highest resolution images of the surfaces of Itokawa, Ryugu, and Bennu were used for their analysis. I don't agree with this. At least, in the case of Itokawa and Ryugu, there must be other images with higher resolution than the images of Figs. 2b and 2c. Moreover, I doubt that the images selected by the authors (Figs. 2b, 2c and 2d) are the representative of the three asteroid surfaces. Please discuss the bias of the observation points in detail.

To resolve the issue on the bias of the chosen observation points that occurred because only one image analysed, several images are now analysed for each body. These images have been selected according to clearly defined criteria, with as different geographical locations as possible (different latitudes and longitudes) to have

a global boulder morphology analysis through the body, and not only at one place. See line 188 – 201 (Methods – image selection), and Figure 1.

(4) The authors estimated the internal friction angle based on an empirical equation (eq. (11)) obtained from laboratory experiments on the Earth using the average roundness of the particles. However, since the gravity on small asteroids is very different from that on the Earth, I doubt that the empirical equation is applicable to low-gravity asteroids. The authors should explicitly discuss the relationship between the empirical equation and the gravity.

The internal angle of friction is parameter inherent to the material that is non-gravity dependant and determined from the Mohr Coloumb law. It is characterised in triaxial shear tests (measuring when a material fails when sheared for a given normal stress), along with the material cohesion. The angle of repose, however (which is often used as a proxy for the internal angle of friction) has indeed been shown to be gravity dependant. Cohesive forces and reduced normal forces (and thus reduced friction) can lead to variations in the angle of repose with gravity.

The following text has now been added to the paper in the Methods.

“This relationship has been obtained in laboratory experiments studying several types of sands, crushed sands and glass beads. The roundness of the particles was computed from binarized 2D images with a MATLAB code\cite{zheng_traditional_2015}. The internal friction angles (ϕ) of these different soils were measured using a triaxial shear test. The experiment measure under which shear stress a material fails for a given normal stress. The internal angle of friction (along with the material cohesion) is then determined from the Mohr-Coulomb la. Given that these parameters are intrinsic to the material, the results are not gravity dependant, unlike the angle of repose which can vary under different gravitational conditions \cite{kleinhans_static_2011}.”

Specific comments

Fig. 1, “rubble pile asteroids...”

Typo? "Rubble pile asteroids..."

Typo has been corrected, thank you. See fig 1.

Fig. 1c, d, “The yellow square indicates the geographical location of the highest resolution image that we used acquired of the surface of each of these asteroids.” The yellow squares do not correspond to the images shown in Fig. 2. Please specify the location.

All locations (longitude and latitude) of the images are now referred to in Table 2. These values are from the PDS. The images in Fig 1 has been modified accordingly.

Fig. 2, Why is there no image number listed?

Images numbers/file names are now all listed in table 2.

L67, "... are elongated (width-to-length ratio of 0.60 ± 0.16) and ..."
As described before, it is possible that the width-to-length ratio of 0.60 is small due to the definition of the boulder contour excluding its shadow. Please discuss the relationship between the boulder contour and the shadow.

The apparent aspect ratio is now of 0.69 after the new analysis with the new boulder semi-automatized boulder selection. (Table 1.)

L76,

It is strange to see that Fig. 4 is placed before Fig. 3.

Corrected

L79-81,

I think the authors have misunderstood these papers (35,36,37). For example, Michikami et al. (2022) claimed that, almost all of the boulders on Eros are the products of impact cratering, while most of the boulders on Ryugu and Itokawa are primarily products of the catastrophic disruption (section 4.2 of their paper).

Thank you for pointing this out. We apologise for the confusing phrasing regarding Eros. We have now made changes regarding those sentences.

Lines 83-95: Fragments formed during laboratory impact experiments typically have an average axial ratio (b/a) of 0.70-0.74. Moreover, fragments formed by catastrophic disruptions are more oblate ($c/a \approx 0.5$) while fragments formed by impact cratering are flatter ($c/a \approx 0.2$)\cite{michikami_fragment_2016}. Previous work has relied on such experiments to conclude that most boulders of diameters ranging from 5 to 85 m on Itokawa and Ryugu\cite{michikami_shape_2010, michikami_boulder_2019, michikami_boulder_2021}, have likely been formed by catastrophic disruption based on the observed axial ratios of the surface boulders and boulders on Eros were formed by impact cratering\cite{michikami_boulder_2021}. In these studies, the axial ratios, obtained by fitting ellipses on the boulders observed at Ryugu and Itokawa surfaces, were found to be slightly smaller than in the laboratory experiments (0.68 and 0.63 respectively) whereas boulders at the surface of Eros were less elongated (ratio of 0.72)\cite{michikami_boulder_2021}. However, the observed discrepancy between the ellipsoidal ratios and the laboratory impact ratio is attributed to the inclination of boulders on asteroid surfaces. For Itokawa and Ryugu, where the gravity is lower and then does not force most boulders to lie on their c axes perpendicular to the asteroid surface. This inclination influences the apparent axial ratio. For Eros, the gravity is relatively high, i.e. the c axes of the boulders is most likely to be perpendicular to the surface and thus the apparent axial ratio is closer to the actual axial ratio\cite{michikami_boulder_2021}.

Tatsuhiro Michikami, Akiko M. Nakamura, and Naru Hirata. The shape distribution of boulders on Asteroid 25143 Itokawa: Comparison with fragments from impact experiments. *Icarus*, 207(1):277–284, May 2010.

Tatsuhiro Michikami, Axel Hagermann, Tokiyuki Kadokawa, Akifumi Yoshida, Akira Shimada, Sunao Hasegawa, and Akira Tsuchiyama. Fragment shapes in impact experiments ranging from cratering to catastrophic disruption. *Icarus*, 264:316–330, January 2016.

Tatsuhiro Michikami, Chikatoshi Honda, Hideaki Miyamoto, Masatoshi Hirabayashi, Axel Hagermann, Terunori Irie, Keita Nomura, Carolyn M. Ernst, Masaki Kawamura, Kiichi Sugimoto, Eri Tatsumi, Tomokatsu Morota, Naru Hirata, Takaaki Noguchi, Yuichiro Cho, Shingo Kameda, Toru Kouyama, Yasuhiro Yokota, Rina Noguchi, Masahiko Hayakawa, Naoyuki Hirata, Rie Honda, Moe Matsuoka, Naoya Sakatani, Hidehiko Suzuki, Manabu Yamada, Kazuo Yoshioka, Hirotaka Sawada, Ryodo Hemmi, Hiroshi Kikuchi, Kazunori Ogawa, Seiichiro Watanabe, Satoshi Tanaka, Makoto Yoshikawa, Yuichi Tsuda, and Seiji Sugita. Boulder size and shape distributions on asteroid Ryugu. *Icarus*, 331:179–191, October 2019.

Tatsuhiro Michikami and Axel Hagermann. Boulder sizes and shapes on asteroids: A comparative study of Eros, Itokawa and Ryugu. *Icarus*, 357:114282, March 2021

Fig. 3, The authors showed the kernel density estimation in Fig. 3. The value of the width-to-length ratio must be between 0 and 1. However, the kernel density estimation includes the value greater than 1. In this regard, it is strange to use the kernel density estimation in principle. Anyway, please clearly explain the value greater than 1. In addition, please clearly define “the bin of the width-to-length ratio”.

We add an explanation in the methods for the kernel density estimate.

Lines 315-317 (in Methods – Kernel Density Estimate): In our case, it smooths the histograms of our data in order to compare them more effectively (For example, see figure 4). This smoothing is the reason why values can exceed 1 on figure 4 even if no value exceeds 1 in the dataset.

L106-108,

Why didn't the author investigate the boulders on the close-up images of Eros? If the authors examine the morphology of the boulders on Eros, more useful information might be obtained in combination with the data of Fig. 4.

It would be indeed interesting, however for this study, we decided to focus on rubble pile asteroids, we hope we can extend our analyses to other bodies in the future. In addition, the images from Eros do not enable to have clear outline of the boulders making it difficult to have a reliable boulder morphology analysis for this body.

L127, “By comparison with laboratory impact experiments (35;36;37), ...”

These references are not the papers in terms of laboratory impact experiments. Please make sure that references are properly cited in the text.

The correct reference is now the only cited in this phrase.

L218,

Is the denominator of the eq. (10) multiplied N by r_{ins} ? If so, please show it explicitly.

It has been corrected (indeed it was a multiplication), thanks. See equation 7 (line 273).

Reviewer #2 (Remarks to the Author):

This is a neat study of rubble pile asteroids and their mechanical properties via image analyses of surface boulders. The manuscript is well written. Here, I list a few comments that I trust would elevate the study in terms of its readability and reproducibility.

1. Describe how is angle of friction derived

The following text has now been added to the paper in the Methods.

“This relationship has been obtained in laboratory experiments studying several types of sands, crushed sands and glass beads. The roundness of the particles was computed from binarized 2D images with a MATLAB code\cite{zheng_traditional_2015}. The internal friction angles (ϕ) of these different soils were measured using a triaxial shear test. The experiment measure under which shear stress a material fails for a given normal stress. The internal angle of friction (along with the material cohesion) is then determined from the Mohr-Coulomb la. Given that these parameters are intrinsic to the material, the results are not gravity dependant, unlike the angle of repose which can vary under different gravitational conditions \cite{kleinhans_static_2011}.”

J. Zheng and R. D. Hryciw. Traditional soil particle sphericity, roundness and surface roughness by computational geometry. *Géotechnique*, 65(6):494–506, June 2015.

M. G. Kleinhans, H. Markies, S. J. de Vet, A. C. in 't Veld, and F. N. Postema. Static and dynamic angles of repose in loose granular materials under reduced gravity. *Journal of Geophysical Research: Planets*, 116(E11), November 2011. Publisher: John Wiley & Sons, Ltd.

2. You describe a pipeline for high res image analysis. Will you make this code publicly available as part of data repository? It is hard to judge its product without the code.

We now provide all the images, files with the boulder contours, output data files and codes to make all of the figures.

3. Some boulders were hand-picked and checked against the average properties. This is fine, but there is no discussion how are boulders picked from the images overall in the complete dataset.

The boulders were not hand-picked, and we changed a little bit the method because we now use several images for Itokawa, Ryugu and Bennu. We analyse boulders larger than 30 pixels.

Lines 285-301 (in Methods – Boulder sample selection for smaller scale descriptors)

Could you discuss the limitations in using the selected high resolution images?

We had limitation regarding phase angle, emission angle as well as representativity of the asteroid but now we use several images for Itokawa, Ryugu and Bennu with criteria on phase angle, resolution and location when it's possible.

Lines 188 – 201 (in Methods – Image selection)

Any other bias coming from using humans instead of ML methods, etc. Have you thought about applying pattern recognition methods (this may be beyond the scope, but still worth mentioning)

To avoid human subjectivity as far as possible, the boulder outlines are now retrieved semi-automatically using a machine learning algorithm (segmentanygrains with python), followed by a human verification in order to confirm each boulder and its outline. See line 202-219 (Methods – Semi automatized image segmentation)

4. Also, how are high resolution images selected? Can that subset of images be compiled in the supplementary materials? If not that, then provide a list of image IDs that can then be pulled from the PDS. This is to insure reproducibility of your work.

We now use a broader dataset with defined criteria for the selected images. The filenames and the criteria are in the paper (Table 2).

5. It is not very clear (at least to me from the first read) what are elements compared with previous studies and what is done in this work beyond the image analyses from DART. Did the team process the other 3 asteroids image sets, or was that data used from elsewhere? I would think that this is all your work here, but not much of it was clarified in the introduction/abstract, hence my confusion.

It has been clarified in the introduction that we analysed the last complete image of Dimorphos captured by DRACO and carefully select images of the surfaces of Itokawa, Ryugu and Bennu.

Reviewer #3 (Remarks to the Author):

This study reports the estimated internal friction angle of various asteroids. The method of estimating internal friction angle is based on an empirical relation between internal friction angle and the average roughness of particles. Various morphological features of the best 30 boulders picked up from asteroids Dimorphos, Itokawa, Ryugu, and Bennu were measured and analysed. As a consequence, this study reveals that the boulder shapes are similar among these bodies. Particularly, the representative value of the internal friction angle (for rubble-pile asteroids) was estimated as 34 ± 6 degrees. This value seems to be almost independent of details of asteroids. This result is informative for those who want to perform numerical simulations of the asteroidal internal structure. In general, the internal properties of asteroids are difficult to estimate. This study provides a possible answer to that problem. Besides, the aspect ratio of measured boulders is also roughly independent of the details of asteroids. The distribution of the aspect ratio is almost universal. The average value suggests the origin of boulders is probably catastrophic impact. The feature particular for asteroid Itokawa is relatively narrow distribution. According to the discussion of this study, the reason for this is the migration-induced segregation of boulders. The analyses seem to be carried out very carefully and the results obtained are convincing.

However, at the same time, the reported results are not very surprising. Therefore, I think this paper deserves publication in more specialized journals. While this manuscript is well written, I have some (relatively minor) comments as listed below.

We would argue that our analysis provides an important measurement technique for determining the angle of internal friction of planetary surface material that has never been applied before on asteroids. We note that our results are consistent with other completely independent analyses of Dimorphos [Barnouin & al., 2024, Pajola & al., 2024], thus highlighting the value of this image-based analysis.

Olivier Barnouin, Ronald Ballouz, Simone Marchi, Jean-Baptiste Vincent, Harrison Agrusa, Yun Zhang, Carolyn Ernst, Maurizio Pajola, Filippo Tusberty, Alice Lucchetti, Ronald Daly, Eric Palmer, Kevin Walsh, Patrick Michel, Jessica Sunshine, Tony Farnham, Derek Richardson, Laura Parro, Naomi Murdoch, and Andrew Rivkin. The geology and evolution of a the Near-Earth binary asteroid system (65803) Didymos. October 2023.

M. Pajola, O. S. Barnouin, A. Lucchetti, M. Hirabayashi, R.-L. Ballouz, E. Asphaug, C. M. Ernst, V. Della Corte, T. Farnham, G. Poggiali, J. M. Sunshine, E. Mazzotta Epifani, N. Murdoch, S. Ieva, S. R. Schwartz, S. Ivanovski, J. M. Trigo-Rodriguez, A. Rossi, N. L. Chabot, A. Zinzi, A. Rivkin, J. R. Brucato, P. Michel, G. Cremonese, E. Dotto, M. Amoroso, I. Bertini, A. Capannolo, A. Cheng, B. Cotugno, M. Dall'Ora, R. T. Daly, V. Di Tana, J. D. P. Deshapriya, I. Gai, P. H. A. Hasselmann, G. Impresario, M. Lavagna,

A. Meneghin, F. Miglioretti, D. Modenini, P. Palumbo, D. Perna, S. Pirrotta, E. Simioni, S. Simonetti, P. Tortora, M. Zannoni, and G. Zanotti. The boulder size-frequency distribution of a binary NEA: a Didymo origin for Dimorphos accreted blocks. Nature Communication, page in review, 2023

1. This paper basically insists the difference among the focused asteroids is almost negligible. The estimated average value of internal friction is close to the usual granular one. In other words, the result is natural and researchers have assumed more or less similar internal friction value. If the difference between different asteroids can be quantitatively revealed by this analysis, it must be very interesting. However, the current measurement uncertainty prevents us from such precise discussion. In this sense, the result is not very surprising. More precise measurements are expected as future problems. The required measurement precision to find the difference among the asteroids should be discussed more quantitatively (based on the current measurement results), if possible.

The angles of internal friction in the paper are derived from the values of roundness. To make sure our analysis pipeline doesn't always give the same roundness values we analysed "extreme" cases i.e., very rounded and very sharp shapes in order to show that the pipeline can indeed provides trustful results regarding the morphology of boulders at the surface of asteroids.

See Methods - Determination of morphological parameters, lines 289-296

Note that it is interesting that the asteroids that appear different actually have boulders with similar morphologies.

2. From asteroidal surface terrain, the distribution of local slopes relative to the local horizontal plane can be measured. From this method, a typical angle (of repose) of Ryugu was estimated as 31 degrees (Watanabe et al. Science 364, 268 (2019), Supplementary Material). Although the internal friction angle and angle of repose are different quantities, they usually show similar values, and this is the case in this study, too. Some discussion about the relation between some friction angles should be discussed.

The internal angle of friction is a non-gravity dependent parameter that is determined from the Mohr Coulomb law. It is characterized in triaxial shear tests (measuring when a material fails when sheared for a given normal stress), along with the material cohesion. The angle of repose, however (which is often used as a proxy for the internal angle of friction) has indeed been shown to be gravity dependent. Cohesive forces and reduced normal forces (and thus reduced friction) can lead to variations in the angle of repose with gravity.

The following text has now been added to the paper in the Methods.

“This relationship has been obtained in laboratory experiments studying several types of sands, crushed sands and glass beads. The roundness of the particles was computed from binarized 2D images with a MATLAB code\cite{zheng_traditional_2015}. The internal friction angles (ϕ) of these different soils were measured using a triaxial shear test. The experiment measure under which shear stress a material fails for a given normal stress. The internal angle of friction (along with the material cohesion) is then determined from the Mohr-Coulomb la. Given that these parameters are intrinsic to the material, the results are not gravity dependant, unlike the angle of repose which can vary under different gravitational conditions (kleinhans_static_2011).”

3. Actually, the form Eq.(11) simply interpolates two extreme values 20 degrees and 45 degrees by using a parameter R. The plot of $\phi(R)$ and 4 values corresponding to each asteroid should be presented. Such a plot visually indicates how close the roundness of asteroids, is directly.

We have made this figure with the bivariate kernel density to see the distributions of both roundness and resulting internal friction angle and added it to the Methods, figure 14.

4. In Fig. 11, the distribution of Ryugu (with 30 best-resolved) shows a bimodal tendency. This means that the typical boulders are not well resolved. The reason why this study can neglect this tendency should be clearly mentioned.

This is a tendency we do not observe anymore as we now analyse more images. However, we could probably have neglected this tendency because the average values in both cases were similar, and we were more interested in the average values rather than the distributions.

REVIEWERS' COMMENTS

Reviewer #1 (Remarks to the Author):

I am satisfied with most of the authors' responses and revisions and would like to thank the authors.

I have two comments.

(1) In the revised text, the authors used semi-automatic image segmentation to detect the contours of the boulders. As a result, the number of counted boulders for each image was reduced compared to the previous one, because some boulders could not be identified with this method. Although the authors said that some boulders were discarded when a doubt existed, I think that the authors should clearly describe the difference between a boulder that the semi-automatic image segmentation could recognize and a boulder that it could not.

(2) Do you mean that Table 2 mentioned in the response to reviewers is Supplementary Table 1?

Reviewer #3 (Remarks to the Author):

The authors improved the analysis methods by using semi-automatized image segmentation and carefully performed the boulder shape analysis. Then, they confirmed the conclusion was not affected by the detailed analysis methods. All these rubble pile small bodies have similar roundness of boulders. This implies the internal friction angle is almost identical. Indeed, it might be interesting that all these bodies have similar boulder roundness while they look different. However, I still feel this is not very surprising as long as they are similar rubble pile bodies. This result is rather natural. But this is my feeling. Perhaps, I may evaluate this result too strictly. Basically, this paper is well-written and the conclusion is surely supported by the analyzed results. All the concerns I raised were solved in this revision. In this sense, I recommend the publication of this study although I am not sure this journal is the appropriate venue or not.

REVIEWERS' COMMENTS

Reviewer #1 (Remarks to the Author):

I am satisfied with most of the authors' responses and revisions and would like to thank the authors.

Thank you for the comments, which were very helpful in improving the paper.

I have two comments.

(1) In the revised text, the authors used semi-automatic image segmentation to detect the contours of the boulders. As a result, the number of counted boulders for each image was reduced compared to the previous one, because some boulders could not be identified with this method. Although the authors said that some boulders were discarded when a doubt existed, I think that the authors should clearly describe the difference between a boulder that the semi-automatic image segmentation could recognize and a boulder that it could not.

The number of boulders identified with the first step segmentation with the machine learning method were not significantly smaller than the previous method. The reason why the number of boulders per image is less now than the previous method, is that we have implemented stricter selection criteria (not cut-off from shadows or buried under another boulder). As a consequence, more boulders are now discarded increasing the reliability of our results.

The following has been slightly modified: (Third paragraph in the *Method - Semi-automatic image segmentation* section)

This semi-automatic method enables to process more images in a smaller amount of time than if the images were segmented completely manually and therefore allows to analyse more areas on asteroids. Even if more boulders could be identified manually on a single image, this method decreases the biases from human interpretation in the boulder outline definition.

(2) Do you mean that Table 2 mentioned in the response to reviewers is Supplementary Table 1?

Yes, table 2 is the Supplementary Table 1, sorry for the confusion.

Reviewer #3 (Remarks to the Author):

The authors improved the analysis methods by using semi-automatized image segmentation and carefully performed the boulder shape analysis. Then, they confirmed the conclusion was not affected by the detailed analysis methods. All these rubble pile small bodies have similar roundness of boulders. This implies the internal friction angle is almost identical. Indeed, it might be interesting that all these bodies have similar boulder roundness while they look different. However, I still feel this is not very surprising as long as they are similar rubble pile bodies. This result is rather natural. But this is my feeling. Perhaps, I may evaluate this result too strictly. Basically, this paper is well-written and the conclusion is surely supported by the analyzed

results. All the concerns I raised were solved in this revision. In this sense, I recommend the publication of this study although I am not sure this journal is the appropriate venue or not.

Thank you for the kind comments on the paper. Regarding if the journal is appropriate or not, the paper will be released in a planetary defence special issue hence the submission to Nature Communications.